# WAVELET GPT: WAVELET INSPIRED LLMS

## ABSTRACT

Large Language Models (LLMs) have ushered in a new wave of artificial intelligence advancements impacting every scientific field and discipline. We live in a world where most of the data around us, e.g., text, audio, and music, has a multi-scale structure. This paper infuses LLMs with a traditional signal processing idea, namely wavelets, during pre-training to take advantage of the structure. Without adding **any extra parameters** to a GPT-style LLM architecture in an academic setup, we achieve the same pre-training performance almost twice as fast in text, audio, and images, by imposing a structure on intermediate embeddings. When trained for the same number of training steps, we achieve significant gains, comparable to pre-training a larger neural architecture. Further, we show this extends to the Long Range Arena benchmark and several input representations such as characters, BPE tokens, bytes, waveform, math expression, image pixels. Our architecture allows every next token prediction access to intermediate embeddings at different temporal resolutions in every decoder block. We hope this will pave the way for incorporating multi-rate signal processing instead of going after scale.

## 1 INTRODUCTION AND RELATED WORK

LLMs have ushered in a super-renaissance of AI advancements and are touching every scientific and engineering discipline. At the heart of this is the Transformer architecture (Vaswani et al., 2017), initially proposed for machine translation. Transformer architecture became the backbone of GPT (Generative Pretrained Transformer) language models (Brown et, 2020) first proposed by Open-AI. Modern LLMs are trained on a straightforward objective: To predict the next token given the previous context, preserving the causality. This not only works for language but also for robotics (Brohan et al., 2023b;a), protein sequences (Madani et al., 2020), raw audio waveforms(Verma & Chafe, 2021), acoustic/music tokens (Huang et al., 2019; Verma & Smith, 2020; Borsos et al., 2023), videos (Yan et al., 2021) etc. This simple recipe of tokenization/creating an embedding and feeding it to transformers also has given rise to non-causal architectures such as BERT(Devlin et al., 2019), Vision Transformers (Dosovitskiy et al., 2021), Audio Transformers (Verma & Berger, 2021) and Video Transformers (Selva et al., 2023). With increased performance by scale, LLMs are reaching hundreds of billions to trillions of parameters (Brown et, 2020; Fedus et al., 2022).

Recent concerns suggest AI research is shifting from academia to industry, according to a Washington Post article by (Nix, 2024). This work aims to enhance LLM capabilities to match those of larger architectures or achieve equivalent performance in fewer training steps. Knowledge distillation (Hinton et al., 2015), uses a larger model to guide a smaller one. (Gu et al., 2024) used KL divergence to enhance next-token prediction from teacher model feedback model rather than training the smaller one from scratch. Model pruning (Sun et al., 2024) removes weights to match the same performance as a large model like LLAMA (Touvron et al., 2023), with fewer compute flops during inference, still relying on a larger model. Dettmers et al. (2024) focus on improving inference or fine-tuning existing models. Unlike distillation and pruning our approach focuses on improving performance during pre-training from scratch. (Nawrot et al., 2022), proposed hierarchical transformers using upsampling-downsampling operations achieve results comparable to those of Transformers but with more efficient computation. Clockwork RNN (Koutnik et al., 2014) improves long-context modelling by splitting RNN neurons into modules that update at different clock rates. Only a few modules activate at each time step. Our approach modifies intermediate embeddings with simple tweaks without using separate learning modules or varying update rates.

Tinkering with the intermediate embeddings: Tamkin et. al (2020) proposed hand-tuned filters on the Discrete Cosine Transform- DCT (Ahmed et al., 1974) of the latent space for different NLP

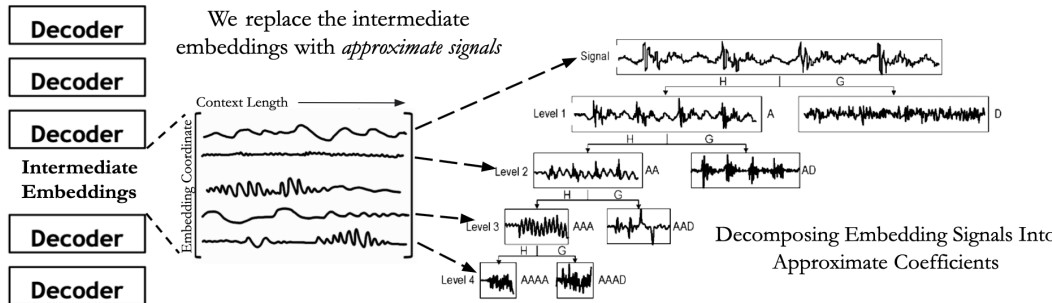

Note: H - Low pass filter; G - High pass filter; A - Approximate information; D - Detailed information

Figure 1: Manipulating signals between GPT decoder blocks by computing 1-D causal discrete haar wavelet transform/learnable approximation at different levels capturing multi-scale structure for each signal. (Right) From Gao & Yan (2006) explaining non-stationary signal processing for signals. Leftmost route of approximate coefficients to model coarsest to finest scales.

tasks for non-causal BERT (Devlin et al., 2019). Computing DCT over context length makes it not applicable for causal architectures like LLMs. There has been work on applying signal processing to BERT-like non-causal architectures. We discuss two here, FNet and WavSPA. They focus on improving attention block, which differs from our work on GPT, which retains a vanilla attention layer. FNet proposed by Lee-Thorp et al. (2022) removes the costly attention mechanism, replacing it with a 2-D FFT block. This operation is non-causal as it looks into future tokens for computing 2-D FFT. WavSpA (Zhuang et al., 2024) carries attention mechanism in the wavelet space. The input sequences are transformed into wavelet space, and the attention mechanism is carried out and then reconstructed. However, computing wavelet transform is non-causal, making them non-applicable for GPT-based LLMs as they look at the entire sequence length (Fig 1 (Zhuang et al., 2024)).

Our work is inspired by neuroscience, which provides evidence that human brain learns multi-scale representations for language at multiple time scales (Caucheteux et al., 2023) instead of fixed-resolution representations. We impose multi-scale representation onto every intermediate decoder embedding at different dimensions. To the best of our knowledge, the paper's contributions are: 1) We propose the first instance of incorporating wavelets into LLM pre-training. We add multi-scale filters onto each of the intermediate embeddings of decoder layers using the Haar/learnable wavelet pipeline. This allows every next token prediction access to multi-scale intermediate embeddings instead of being fixed-resolution in every decoder layer representation. 2) We show speedups in pre-training of GPT, like transformer-based LLM in the range of 40-60%, with adding a multi-scale structure. With same training steps, the model gives a performance boost akin to adding more layers.

## 2 DATASET

We use four open-source datasets from natural language, symbolic music, speech tokens, and raw audio waveform for next token prediction. For text, we choose text-8 (Mikolov et al., 2012). We choose this over other datasets as i)it is a famous and widely cited character-level language modelling dataset, and ii) it uses a simple vocabulary (space + 26 lowercase characters) to detach the effects of various tokenizers. It has 100M characters with split training split as given by Al-Rfou et al. (2019). For raw audio, the goal is to predict the next sample given the context. We use the YouTube-Mix-8 dataset for long-context modeling (Goel et al., 2022; Verma, 2022). Our vocabulary size is 256, with a sampling rate 16KHz as input is 8-bit. We use a third dataset, MAESTRO (Hawthorne et al., 2019), containing over 1000 MIDI files of classical music pieces with a tokenizer proposed by Huang et al. (2019), which converts MIDI tracks into discrete tokens with a vocabulary size of 388. Finally, we use 1000 hours of LibriSpeech dataset and a widely used ENCODEC Défossez et al. (2022) tokenizer in a setup similar to VALL-E Wang et al. (2023) to model acoustic tokens [1]. The goal in all four modalities is not to chase state-of-the-art pre-training performance, as *this paper was written in an academic setting with very few computational resources*. We show how the model performs in pre-training instead of post-training, as the goal is to build better foundational architectures with the same parameters, pushing the capabilities of smaller decoder architectures.

[1]The goal here is to model the coarsest tokens, as errors in modelling the coarser tokens will lead to the finer tokens being modelled incorrectly as they are conditioned on the coarsest token as shown in VALL-E paper

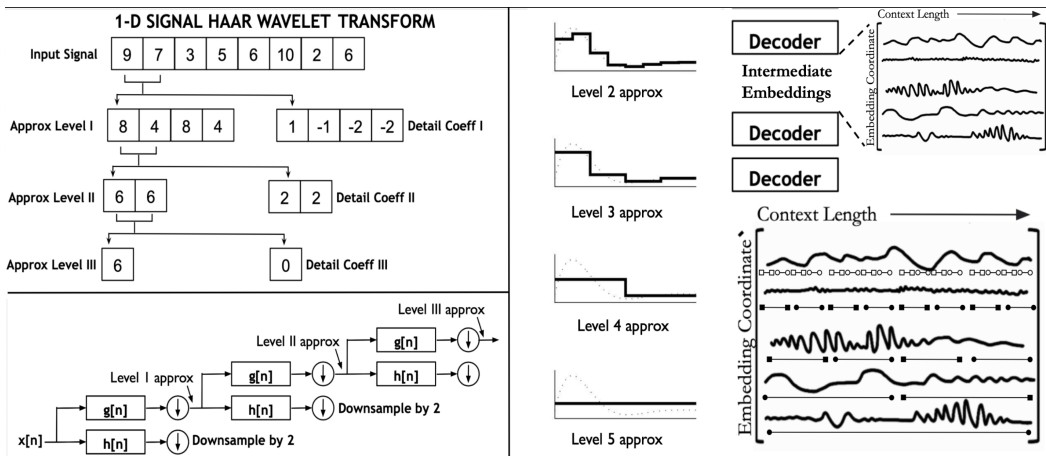

Figure 2: (Bottom L): A 3-level filter bank tree generates signals at different resolutions. Approximate coefficients are computed by applying a wavelet's impulse response & recursively downsampling. (Top L): Approximate and detailed coefficients are iteratively calculated via first-order averages/differences and down-sampling until a single scalar represents the signal. (R): For a 32-length signal, Haar wavelet captures coarsest to finest approximations and is redrawn from (Flores-Mangas, 2014). Embeddings evolve at different rates via causal wavelet approximation, with coarse (level 5) and fine (level 2) resolutions, embedding multi-scale information

## 3 METHODOLOGY

This section will describe the approach to incorporating wavelet inspired computation into transformer-based Large Language Models while retaining causality. The ideas described here are generic as we tinker intermediate embeddings. Thus theycan be easily extrapolated to non-Transformer architectures, e.g. state space model. For any GPT signal, we compute a version of the discrete wavelet transform and incorporate it back into the signal. Let $x_{(i)}^l$ be the output of the $l^{th}$ decoder layer, representing the activation along the $i^{th}$ coordinate, with a dimension equal to the context length $L$ of the transformer-based GPT model. In the original GPT architecture with $N + 1$ layers and embedding dimension $E$, we obtain $N \cdot E$ signals of length $L$ from intermediate embeddings between decoder blocks, where $E$ ranges from $[0 - 128)$ dimensions. For any signal $x[n]$, the discrete wavelet transform resembles passing the signal through filters of varying resolutions, as illustrated in Figure 2. We will use the Haar wavelet, a family of square-shaped functions this paper obtained from a mother wavelet via scaling and shifting operations. Given a mother wavelet function $\psi$, the child wavelets as $\psi_{j,k}[n]$, where $j$ is the scaling factor and $k$ is the shift factor. The relation is given by $\psi_{j,k}[n] = \frac{1}{\sqrt{2^j}} \psi\left(\frac{n - k2^j}{2^j}\right)$. These signals are shifted and scaled to capture information at various time scales, with $n$ representing time or the context length. This concept resembles the diagram in Figure 1, which illustrates capturing different signals in the intermediate layers of Transformer decoders at various resolutions. Discrete wavelet transform, which passes any signal through filters and downsampling operations. This process, shown in Figure 2, is similar to a convolutional neural network (CNN) like ResNet (He et. al, 2016), featuring learned convolutional filters analogous to $h[n]$ and $g[n]$, along with downsampling, such as max pooling. In convolutional architectures, we follow one branch of Figure 2, recursively taking the output of filters and downsampling. This similarity contributed to popularity of wavelets in the 1990/2000s for image understanding, reflecting parallels with convolutional architectures (Huang & Aviyente, 2008; Kingsbury & Magarey, 1998). For Haar wavelets, this is passing the signal through low-pass and high-pass filters corresponding to the kernels $g[n]$ and $h[n]$. The Haar wavelet transform averages and computes differences, with impulse responses $g[n] = \left[\frac{1}{2}, \frac{1}{2}\right]$ and $h[n] = \left[\frac{1}{2}, -\frac{1}{2}\right]$. Figure 2 provides a detailed explanation of the discrete wavelet transform. For a 1-D signal $x[n]$ of length $L$, we get level 1 coefficients by filters $g[n]$ and $h[n]$, followed by downsampling. Thus, the approximation coefficients $y_{\text{approx}}$ and $y_{\text{detail}}$ result from a LTI system defined by convolution followed by downsampling by two (Equation 2). This is seen in Algorithm 1 with type $\in \{\text{approx}, \text{detail}\}$ and $f_{\text{approx}} = g$, $f_{\text{detail}} = h$. The relation is given by $y_{\text{type}}[n] = \sum_{k=-\infty}^{\infty} x[k] f_{\text{type}}[2n - k]$. To obtain multi-scale representations of the original signal, the operation for $x[n]$ is recursively applied to $y_a$

(approx) to derive level 2 wavelet coefficients $y_a^2$ and $y_d^2$ (detail). Here, $x[n]$ represents intermediate signals across the context length at each decoder block output in the LLM. The approximate coefficients $y_a$ and $y_d$, along with their decompositions $\{y_a, y_d, y_a^2, y_a^3, y_a^4, \ldots\}$, are used for further processing. Notably, $y_a^2, y_a^3, y_a^4$ have lengths reduced by factors of $2, 4, 8, \ldots$. The Haar wavelet transform averages adjacent samples while preserving causality by averaging current and past samples. Higher-order coefficients capture averages over larger context lengths, as shown in Figure 2. We can continue until only a single scalar value remains, representing the mean of the signal. The Haar wavelet transform computes averages and differences to create a multi-resolution representation, capturing low and high frequencies at different resolutions. Figure 2 illustrates the same signal captured at coarser and finer representations using Haar wavelets, applied to intermediate embeddings, allowing each next token prediction access to these representations. For the case of learnable wavelet kernels, we create a multi-resolution representation by varying the kernel size (Algorithm 1) to allow the LLM to learn the optimal kernels optimized for the next token prediction.

---

**Algorithm 1** Wavelet-GPT

---

$E$: Model or Embedding Dimension
$L$: Context Length
$N + 1$: Number of Decoder Layers
**for** layer $l = 1, 2, \ldots, N$ **do**
 $\mathbf{x}^l \leftarrow$ Output of the $l$-th Decoder, dimension: $E \times L$
 $\mathbf{xn}^l \leftarrow$ Modified decoder embedding replacing $\mathbf{x}^l$
 $\mathbf{xn}^l_{(i)} \leftarrow \mathbf{x}^l_{(i)}$ for embedding dimension $\qquad\qquad\qquad\qquad\qquad\qquad\qquad\qquad i < E/2$
 $\mathbf{f(i)} \leftarrow 2^F \quad F = int(L_k * (i - E/2)/(E/2 - 1))$
 $L_k = \lfloor \log_2(L) \rfloor + 1 \qquad i \geq \frac{E}{2}$ //Kernel length function of embd coordinate power of 2
 $\mathbf{xn}^l_{(i)}(\mathbf{k}) \leftarrow \frac{1}{\mathbf{f(i)}} \sum_{\mathbf{m=k-f(i)}}^{\mathbf{k}} \mathbf{x}^l_{(i)}(\mathbf{m}) \qquad i \geq \frac{E}{2}$ // for non-learnable fixed Haar wavelet
 $\mathbf{xn}^l_{(i)}(\mathbf{k}) \leftarrow \sum_{\mathbf{m=0}}^{\mathbf{f(i)-1}} \mathbf{h(m)} \cdot \mathbf{x}^l_{(i)}(\mathbf{k - m}) \qquad i \geq \frac{E}{2}$ // for learnable wavelet kernel $h$
**end for**

---

### 3.1 CONNECTING WAVELETS AND LLM EMBEDDINGS

In many signal processing tasks, first-order detail and approximate coefficients capture signals at multiple levels. We apply the same idea to intermediate transformer embeddings across tokens. Real-world data is naturally hierarchical—text spans letters to topics, music from notes to motifs, and speech from phonemes to phrases. With the Haar wavelet, this hierarchy reduces to simple averaging, while in the learnable case, kernel weights are optimized for next-token prediction. Continuing with approximations eventually yields a single scalar—the global average for Haar. To match the original sequence length, approximation coefficients can be expanded, e.g., via up-sampling. We call the length-matched version the approximate signal, distinct from shorter coefficients. In Figure 2 (R), we show this process: applying the kernel at each level (e.g.,$[1, 1]$, $[1, 1, 1, 1]$, etc.). reconstructs multi-scale approximations aligned with the input $x[n]$. This piecewise constant function is shown in Figure 2. LLM embedding coordinates define unique resolution kernels, each corresponding to a specific scale of data. The reconstructed signal $x_{\text{recon}}[n]$, a method to derive the *approximate signal*, is computed from wavelet coefficients $c_j$ at level $j$ as: $x_{\text{recon}}^j[n] = \sum_k c_k \cdot \psi_{j,k}[n]$. Equation 3 requires storing child wavelets at various approximations, complicating the process and rendering it non-causal as computing $c_k$ considers the entire signal. As $c_k$ depends on future information, we cannot use this to reconstruct the signal from its approximate coefficients. To adapt this for LLMs, we simplify the computation of the approximate signal in a differentiable form, extending Equation 3 to both learnable and fixed multi-resolution kernels. For the Haar wavelet, the input is averaged with kernels of increasing length until it approximates the full signal, with kernel size setting the approximation level. Since LLMs assume causality, each location is updated using only prior samples, with left zero-padding when the kernel exceeds the available window. Multi-level wavelet transforms produce signals at different resolutions, which can disrupt intermediate embeddings. We generate resolution-specific signal approximations parameterized by the embedding dimension. In Section 4.4, we make these kernels learnable, allowing the architecture to maintain multi-scale operation (Equation 3), with learnable weights with $x_{\text{recon}}[n]$ learned with varying resolutions.

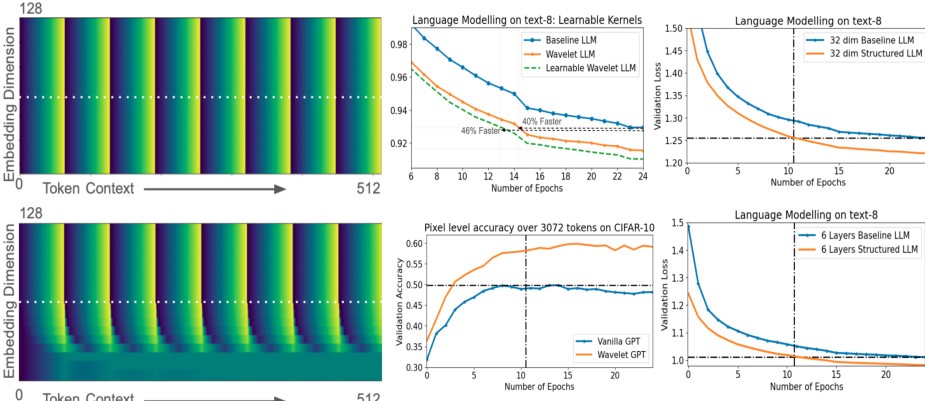

Figure 3: (Left) Toy example showing embeddings before/after imposing multi-rate structure. Different embedding dimensions advance at distinct rates while maintaining causality, as seen from patterns dispersing from dimension 64 to 0. (Right) Validation loss during pre-training on text-8 with learnable multi-scale structure achieving comparable performance nearly twice as fast/performance boost akin to adding additional decoder layers. Our architecture's performance on text-8 with a 32-dim model matches the speedup similar to that seen for 128-dim and shallower models. LRA image benchmark, a 10% performance increase without adding any parameters

### 3.2 WAVELET COEFFICIENTS BY EMBEDDING COORDINATES

One option is to compute *approximate signals* for each coordinate signal $x^l_{(i)}$ across decoder layers at levels I–IX. For a context length of 512, this yields nine signals with resolutions 512, 256, 128, 64, 32, 16, 8, 4, and 2—dramatically increasing complexity and requiring major GPT modifications. We instead propose parameterizing the level by embedding dimension index, avoiding the need to compute all approximations. The goal is to nudge embeddings only slightly toward the inductive biases we impose, without over-constraining what they learn. Since transformers succeed even without biases, our approach seeks the best of both worlds by steering only half the embedding dimensions. We adjust intermediate GPT embeddings in only half the dimensions. Embeddings from 0 to $E/2$ (coordinates 0 to 64 when $E = 128$) remain unchanged. For the rest, we apply processing based on their index $i$. If $x^l(i)$ is an intermediate embedding after the $l^{th}$ decoder layer along the $i^{th}$ dimension, the modified signal $xn^l(i)$ equals $x^l_{(i)}$ for $i \in [0, E/2]$. For $i > E/2$, we impose structure using an approximate signal calculated from wavelet coefficients corresponding to the index $i$. We use a mapping function $f$ that takes coordinate $i$ (ranging from $E/2$ to $E$) and returns the kernel size corresponding to approximation levels from $I$ to $IX$. The linear function gradually increases from level $I$ (kernel size two at $i = E/2$) to level $IX$ (kernel size 512 at $i = E$, or the coarsest representation i.e., a scalar). Now, let us find out how we compute the modified new signal $xn^l_{(i)}$ that replaces the original intermediate Transformer embeddings $x^l_{(i)}$. $f(i)$ is the kernel size for the coordinate $i$. The modified signal is either kept the same or modified as $xn^l_{(i)}(k) = \frac{1}{f(i)} \sum_{m=k-f(i)}^{k} x^l_{(i)}(m)$ as seen in Algorithm 1. For cases where $k - f(i) < 0$, we zero-pad the signal to ensure valid average/kernel computation. Specifically, for the Haar wavelet, the modified signal acts as a causal moving average filter with finite length, averaging the embedding signal along the $i^{th}$ coordinate with a kernel size determined by $f(i)$. This operation does not introduce new parameters or maintain causality in LLMs to prevent future token leakage, as seen in Equation 4. In Algorithm 1, each value of the modified signal at token $k$ is computed using a convolution with a learned kernel $h(.)$ and variable length $f(i)$, parameterized by the embedding coordinate dimension $i$. Each kernel is learned independently for every signal.

### 3.3 IMPOSING STRUCTURE: TOY EXAMPLE

In Figure 3, we illustrate a toy example of how we impose structure onto decoder Transformer embeddings. The left side shows eight variations along the token dimension, with onset/sudden bursts at token indices 32, 64, etc., decreasing to zero before rising again. As discussed in the introduction, datasets inherently possess a hierarchical structure, which we capture by imposing intermediate Transformer embeddings at each layer. In this example, we retain embeddings at the original resolution for half the dimensions (split by a white line). For the other half, we gradually

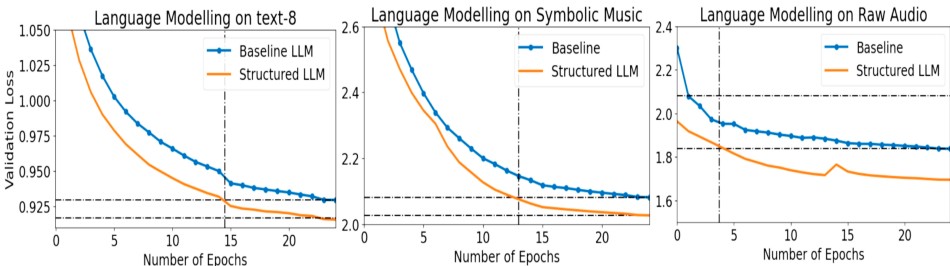

Figure 4: Results for natural language, symbolic music, and raw audio. We perform faster than baseline, almost twice as fast on shrunk-down GPT. We see substantial gains in pre-training performance for the same epochs, equivalent to a much larger architecture. The black vertical line denotes the epoch at which our architecture achieves the same performance as our baseline architecture.

increase the kernel length across the context and compute the average causally. The final embedding dimension averages over the token dimension with a kernel size equal to the context length (zero-padding if necessary). This creates highways, allowing embeddings to move at different rates: the coordinates from $E/2$ to $E$ move at the Transformer's original speed, while those from 0 to $E/2$ transition from faster to slower movement. This approach enables the attention mechanism to utilize multi-scale features at varying rates across all layers and tokens, as explored in the next section. Further, this multi-scale structure can be made learnable, driven by just the next token prediction.

## 4 EXPERIMENTS

The main aim of these experiments is to show that the pre-training performance of the models across four modalities improves with/without doing intermediate modifications on embeddings inspired by wavelets. We also benchmark on LRA tasks. We propose a shrunk-down GPT baseline architecture that has the same topology. We do not compare against larger architectures, as this paper focuses on pre-training from scratch, and was written in with access to limited computational resources in academia. We evaluate pre-training performance with and without wavelet-inspired blocks. We only report how well our generative model does for pretraining by quantifying the likelihood scores similar to papers such as Mega-Byte Yu et al. (2023) and Music Transformer Huang et al. (2019) that only report NLL scores in the entire paper. We also validate our method across various modalities such as text, audio and music. Further, we benchmark it on various input representations such as raw waveform, MIDI tokens, acoustic tokens, text bytes, characters, and BPE tokens, in addition to math expressions. Our experiments, based on the GPT-2 architecture, have 10 Transformer decoder layers with a context length of 512, trained from scratch. Each modality shares the same architecture, using an embedding dimension of 128, a feed-forward dimension of 512, and 8 attention heads. The final decoder outputs a dense layer of 2048 neurons, followed by a layer matching the vocabulary size [2]. Baseline models consist of standard Transformer decoder blocks without modified embeddings. We retain half [3] of the embedding coordinates for our proposed architecture and impose either a fixed or learnable multi-scale structure on the other half for all intermediate layers. All models were trained from scratch in TensorFlow Abadi et al. (2016) for 25 epochs, starting with a learning rate of 3e-4, decreasing to 1e-5 when loss plateaued. Each model utilized 1M training points, totalling 500 million tokens, randomly cropped from the dataset. We measured performance using negative log-likelihood loss, as this method improves the core architecture of the transformer-based GPT - helping achieve the objective we want to achieve: predict the next token correctly. Since we are operating on intermediate embeddings, our work can hopefully generalize to setups with structured data similar to text, raw audio, and symbolic music, where one can go from a fine-grained structure to a coarse structure. We impose a multi-scale structure that allows the attention mechanism to learn dependencies across embeddings and inject some information that can capture coarse and fine-grained structures into embedding coordinates while maintaining causality.

---

[2]Vocab of 27 for text8, 256 for raw waveform (Goel et al., 2022; Verma, 2022), 388 for symbolic music, and 1024 for ENCODEC speech tokens, 256 for 8-bit raw pixels, 50257 for BPE tokens

[3]The choice of half is a hyper-parameter. It is difficult to optimize for every modality/input representation due to computational resource constraints. If the optimal split is not half, it improves already strong results

### 4.1 PERFORMANCE ON MODALITIES

We compared the performance of our baseline architecture across three modalities : text, symbolic music, and audio waveform with and without wavelet-based intermediate operations. Results showed significant performance improvements in all modalities with the same number of training steps. To illustrate, a 0.04 decrease in validation loss is comparable to going from a 16 to a 64-layer model on text-8 dataset (papers-with code, 2024). As shown in Figure 4, our modified GPT architecture achieves this loss nearly twice as quickly in training steps as the original model, showing that GPT-like architecture can take advantage of the structure we imposed on half of the embedding dimensions. This speedup, i.e., the number of epochs/steps taken to achieve the same performance (SP: same performance epoch), is even smaller for raw audio due to the quasi-stationary nature of audio signals at smaller time scales (20-30 ms for harmonic sounds). For a sampling rate of 16KHz, a context length of 512 would correspond to 32ms, which may be one of the reasons that some of the coordinates nail down the contents of the context in fewer coordinates onto which we impose structure. The convergence is significantly faster for the raw waveform LLM setup and achieves nearly twice the speed of text-8 and symbolic music. We also ran benchmarks on LibriSpeech corpus about 1000 hours of speech and acoustic tokens, further strengthening our method is generic to handle several types and modalities of tokens. We run our method on audio classification with Audio Transformer over 200 categories of audio for FSD-50K benchmarks Fonseca et. al (2020). We get a performance boost and speedups, thereby showcasing the ubiquity of our proposed method for generative modelling and classification as well. We also compare the absolute clock run times of our modifications in both learnable/non-learnable setups. Table 1 reports the time to complete one epoch relative to our baseline architecture. Our method is computationally inexpensive, as it only involves fixed kernel multiplication or learning a single filter convolutional kernel with variable context lengths along different coordinates.

| Modality | Base | Ours | SPE | Gain | Time |
|---|---|---|---|---|---|
| Text-8 | 0.93 | 0.92 | 14.5 | 42% | 1.013 |
| Audio | 1.84 | 1.70 | 3.7 | 85% | 1.042 |
| Music | 2.08 | 2.02 | 13 | 48% | 1.059 |
| Text-8(L) | 0.93 | 0.91 | 12.9 | 48% | 1.094 |
| Wiki-103(L) | 4.11 | 4.05 | 9.5 | 62% | 1.130 |
| Speech(L) | 2.43 | 2.40 | 9.2 | 63% | 1.110 |
| FSD-50K | 40.6% | 42.8% | 32 | 65% | 1.037 |

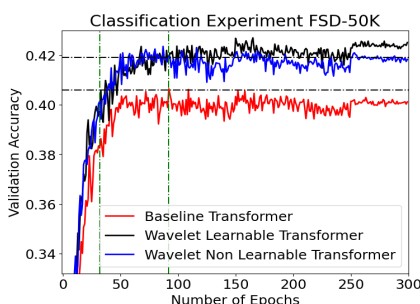

Figure 5: Comparison of the negative-log likelihood (NLL) scores for our architecture across three modalities, with and without wavelet-based fixed/learnable (L) structure. (Left) Table shows the NLL scores and speedup, with Same Performance Epoch (SPE) with baseline as 25 epochs, relative GPU hours. (R) FSD-50K Audio Transformer top-5 accuracy results. Vertical green lines indicate the highest accuracy achieved and the point where the same accuracy is reached 60% faster.

### 4.2 MAKING MULTI-SCALE KERNELS LEARNABLE

We extend our approach by allowing each kernel to be learnable. In the previous section, we defined the kernel shapes and computed approximate signals of intermediate layer activations across all layers, with different resolutions at varying embedding dimensions to emulate a causal wavelet transform. Here, each kernel of length $L$ at a given level is learnable, providing an alternative method to compute the *approximate signal*. By learning the kernel weights, the model can adaptively weight each decoder layer dimension instead of relying on fixed kernels, such as exponentially weighted averages. As outlined in Algorithm 1, this introduces only 20k parameters, or 0.2% of the base decoder architecture. Due to resource constraints, we run experiments for both learnable and fixed kernels, three variants each. Intuitively, an optimal kernel exists whose shape is better suited for next-token prediction for a specific modality than a simple Haar wavelet. This adaptation further improves performance, achieving a speedup from 42% to 48% to reach comparable baseline performance, as shown in Figure 4 on the text-8 dataset. We also benchmark on Wiki-103 using the GPT-2 tokenizer, yielding even larger gains. Figure 5 illustrates that our approach matches the performance of a 10-layer architecture at more than twice the speed. Beyond faster convergence, we observe a 3.6-point improvement in perplexity scores over the baseline for Wiki-103. Section 4.4 demonstrates it scales with model size and depth, showing potential for larger LLM architectures.

### 4.3 Ablation on Depth And Model Dimension

The aim for these experiments was to see if our model scales with depth of the Transformer and the model dimension. We explore two architecture variants on text-8: (i) reducing the model dimension from 128 to 32 and (ii) reducing the number of layers. The model with 32-dimensional, 10 decoder layers (eight heads) achieves baseline performance in around ten epochs and runs nearly twice as fast (Figure 4). For the second experiment, we retain the architecture from Table 1 but reduce the Transformer decoder to six layers while keeping other parameters unchanged (feed-forward dimension four times the model dimension, eight heads). With Haar-inspired modifications, the model matches baseline performance twice as fast, consistent with the results reported in Section 4.1. While it is difficult to scale the architectures beyond a certain depth and model dimension in academic setups: we believe that by seeing the effect of model dimension and depth holding, we are confident that the findings will extrapolate for much larger/deeper models.

### 4.4 Audio Classification Benchmark

We explore the strength of our method for a typical audio classification on a standard audio classification benchmark FSD-50K. The goal is to identify the sound categories in 1s of audio correctly. We use a transformer-based architecture similar to Audio Transformer as our baseline model. It consists of 128 convolutional filters with a length of 200 learned over 25ms of audio sampled at 16KHz, yielding patches of 400-length audio samples. The convolutional filter output is then max-pooled across the 25ms window to give a single vector of length 128, i.e. the number of convolutional filters being fed to a Transformer stack of 6 layers with model dimension as 64 similar to Verma & Berger (2021). We report on the top-5 % accuracy and relative gain in mAP scores for our proposed architecture with learnable/non-learnable kernels. We see that we get a performance boost of about 2% and a faster convergence of more than 60%, shown in Figure 5, for baseline/ learnable/fixed kernels.

### 4.5 Similarities and Differences With EMA

We compare Exponential Moving Averages (EMA) on intermediate signals. Unlike the Haar wavelet, which takes fixed window weights, which takes the mean of the signal in the window, EMA uses an exponential kernel. Let the signal $x_i^l(t)$, after the $l^{th}$ layer, be of length equal to context length, with $t$ being the token index from 0 to $L$ at embedding dimension $i$. The modified signal $s_t$ is: $s_0 = x_i^l(0)$ $s_t = \alpha x_i^l(t) + (1 - \alpha)s_{t-1}$ where $\alpha$, the decay factor, satisfies $0 < \alpha < 1$. Unlike an EMA, our method captures multi-scale information using a finite kernel with zero weights outside a specified length. In text-8 experiments, we applied EMA on half of the embedding dimensions, with $\alpha$ linearly varying between 0 and 1 for dimensions 64 to 128 mimicking our approach with a classical approach. This under-performed compared to our baseline, with an NLL score of 0.94, while our baseline and proposed method achieved scores of 0.93, 0.92, and 0.91 for non-learnable and learnable cases, respectively. Our method provides a simple, signal processing-based scheme that optimizes weights across multiple resolutions driven by next-token prediction and outperforms EMA. Depending on $\alpha$, the EMA filter produces an exponential kernel while we maintain a constant kernel or allow weights learned from scratch optimized for the next token prediction. Further, EMA is an Infinite-Impulse Response (IIR) filter. Consequently, for each value update, the contributions from previous samples never reach zero. These can accumulate significantly at longer context lengths for certain $\alpha$. The recursive, non-learnable nature of the EMA IIR filter ensures some contribution from all embeddings, which explains performance degradation. Our method uses zero weights outside the kernel length, capturing multi-scale information.

## 5 Long Range Arena Benchmarks

We adapt our architecture to the Long-Range Arena (LRA) tasks Tay et al. (2021), evaluating long-range prediction across text, images, and mathematical expressions. These tasks measure the model's ability to capture similarity, structure, and reasoning over extended contexts. Our focus is on Transformer-based architectures, following recent reports (Liu et al., 2024), although other approaches include state-space models, hybrids, or modified attention mechanisms. For text, we perform binary classification on the IMDb review dataset (Maas et al., 2011), using byte-level inputs with a context length of 2048 to predict whether a movie review is positive or negative. For images, we use CIFAR-10 from the LRA benchmark, classifying sequences of 3072 pixels into ten categories. Finally, we benchmark on Long ListOps, which tests the ability to process hierarchically structured data in extended sequences. Our version of ListOps uses sequences up to 2K tokens,

Table 1: Performance on LRA tasks (Tay et al. (2020b)) as reported in Liu et al. (2024). Bold the best-performing model, and underlined indicates the second-best. We use a baseline GPT baseline (Section 5) and modify intermediate embeddings by imposing a hierarchical structure. Non-transformer-based, modified attention-based or hybrid architectures are not reported.

| Attention Based Models | ListOps | Text | Image |
| --- | --- | --- | --- |
| Transformer (Vaswani et al., 2017) | 36.37 | 64.27 | 42.44 |
| Local Attention (Tay et al., 2020b) | 15.82 | 63.98 | 41.46 |
| Linear Trans. (Vyas et al., 2020) | 16.13 | 65.90 | 42.34 |
| Linformer (Wang et al., 2020) | 35.70 | 53.94 | 38.56 |
| Sparse Trans. (Child et al., 2019) | 17.07 | 63.58 | 44.24 |
| Performer (Kaiser et al., 2021) | 18.01 | 65.40 | 42.77 |
| Sinkhorn (Tay et al., 2020a) | 33.67 | 61.20 | 41.23 |
| Longformer (Beltagy et al., 2020) | 35.63 | 64.02 | 40.83 |
| BigBird (Zaheer et al., 2020) | 36.05 | 64.02 | 40.83 |
| Luna-256 (Ma et al., 2021) | 37.25 | 65.78 | 47.86 |
| Reformer (Kitaev et al., 2020) | 37.27 | 56.10 | 38.07 |
| Non-Causal | | | |
| FNET (Lee-Thorp et al., 2022) | 37.27 | 56.10 | 38.07 |
| WavSPA (Zhuang et al., 2024) | 55.40 | **81.60** | 55.58 |
| (Ours) GPT Baseline | 41.65 | 65.32 | 49.81 |
| **(Ours) WaveletGPT** | **57.5** | 66.38 | **59.81** |

requiring the model to access all tokens and capture the logical structure for ten-way classification. This task is particularly challenging due to its hierarchical nature. Data extraction follows the setup of Khalitov et al. (2022), ensuring consistency with other benchmarks. We employ an identical architecture across all three modalities, modifying only the embedding matrix to match the respective tokenizers and output categories. Our baseline is a 6-layer causal Transformer decoder with a model dimension of 32 and a feed-forward dimension four times the embedding size. For classification, we extract the final token as a 32-dimensional embedding, followed by a dense layer of 2048 neurons and a final dense layer matching the number of categories. Inputs are embedded into 32-dimensional vectors, with vocabularies of size 256 for text/image and 16 for ListOps, and context lengths of 2048, 3072, and 1999 tokens, producing 2, 10, and 10 output categories respectively. In our modified architecture, we insert the *waveletGPT* module between each decoder layer, preserving half of the embedding dimensions and applying non-learnable kernels to the other half. These kernels scale linearly from 2, 4, and 8 up to 512 for dimensions 16 to 32, while maintaining causality. This creates hierarchical processing highways at each embedding and Transformer layer without adding parameters, similar to our strategy for pre-trained LLMs. As reported in Table 1, this yields consistent gains across all modalities, with even small improvements being meaningful. We outperform non-causal signal-processing-based approaches, such as (Zhuang et al., 2024), achieving nearly 2% improvement on ListOps and 4.5% on a much smaller architecture (32 dimensions, six layers) compared to theirs (128 dimensions, eight layers). For fairness, we restrict comparisons primarily to vanilla Transformer baselines, but also evaluate two non-causal, signal-processing-inspired architectures: FNet and WavSPA. Relative to non-causal FNet, our model achieves substantial improvements on all three LRA tasks: 20% on ListOps and Image, and 10% on text. The largest gain is observed on ListOps, which requires modeling a hierarchical, tree-like structure, highlighting our model's suitability for such tasks. To the best of our knowledge (Liu et al., 2024), this represents the strongest performance achieved by a simple attention-based Transformer on LRA benchmarks.

# 6 CONCLUSION AND FUTURE WORK

We showcase a powerful integration of a core signal processing idea, wavelets, into large language model pre-training. By imposing a multi-scale structure on every intermediate embedding, we achieve the same performance 40–60% faster than a baseline without adding parameters, and also see a substantial boost when training for the same number of steps. We further demonstrate strong gains on LRA benchmarks by giving each next-token prediction access to multi-scale embeddings in every decoder layer. Our method generalizes across three modalities—raw text, symbolic music, and raw audio—delivering similar speedups on diverse inputs, including raw audio samples, acoustic tokens, MIDI tokens, byte text, math expressions, BPE tokens, raw image pixels, and characters. This highlights its generality for improving pre-training across datasets and input types.

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

## A APPENDIX

You may include other additional sections here.

