# OpenReview forum: "WaveletGPT: Wavelet Inspired LLMs"
_ICLR.cc/2026/Conference — Submitted to ICLR 2026_

### Official Review · Reviewer_79tD · 2025-10-17

**Soundness:** 2
**Presentation:** 1
**Contribution:** 1
**Rating:** 2
**Confidence:** 5

**Summary:**

In this paper, the author incorporate wavelets into GPT-style LLM pretraining by imposing a multiscale structure on intermediate embeddings without adding parameters. They match pretraining quality almost twice as fast on text, audio, and images and, at equal training steps, obtain gains comparable to training a larger model. They also show effectiveness on Long Range Arena and on inputs including characters, BPE, bytes, waveform, math expressions, and pixels, suggesting multi-rate signal processing as a practical alternative to scaling.

**Strengths:**

The idea of incorporating wavelets into GPT may be useful.

**Weaknesses:**

**(1) Outdated and selective baselines.** The abstract claims that “we achieve significant gains, comparable to pre-training a larger neural architecture,” yet most comparisons are against models from three to four years ago. To substantiate the claim, the paper should include strong, up-to-date baselines and parameter-matched ablations with recent open models (e.g., Llama, Qwen, Phi families). As indicated by the title, this is an LLM paper, so comparison with these models—which are becoming the baseline for LLM—is essential. Without such comparisons, the central claim is not convincingly supported.

They describe it as follows in their paper.
>The goal in all four modalities is not to chase state-of-the-art pre-training performance, as this
paper was written in an academic setting with very few computational resources. (L103,104)

Given that the work explicitly targets efficiency at small scale with limited academic compute, the method does not qualify as **Large**LM pre-training in the contemporary sense. The current title/abstract/intro oversell the scope. The paper should adopt small-scale GPT language throughout and revise claims to reflect the actual experimental setting.
Although the paper frames the method as “LLM pre-training,” the actual setup is a shrunk-down GPT: 10 decoder layers with d_model = 128, FFN = 512, 8 heads, context length 512, trained from scratch on small/modest datasets (e.g., text8, MAESTRO, LibriSpeech tokens, YouTube-Mix-8) with 1 M training points ; the authors explicitly state they do not compare against larger architectures and that the work was conducted with very limited compute in academia. Accordingly, the experiments are small-scale and do not constitute LLM pre-training in the contemporary sense.

**(2) Insufficient discussion of prior work and limited novelty positioning.** There is a substantial body of research applying wavelets within Transformer-based models (e.g., for tokenization, attention, positional encodings, or hybrid frequency–time representations). The submission does not adequately survey this literature or clearly articulate its differences. Moreover, the method does not appear to fundamentally modify the GPT architecture to the extent implied by the name “WaveletGPT.” The paper should cite the relevant wavelet-Transformer works and provide a precise comparison that clarifies what is genuinely new.

> To the best of our knowledge, the paper’s contributions are: 1) We propose the first instance of incorporating wavelets into LLM pre-training. (L83,84)

The statement  is clearly erroneous. It is simply a result of the authors' insufficient survey. Even a quick search of ICLR papers already surfaces multiple works that integrate wavelets into Transformer components. For example, Adaptive Wavelet Transformer Network (ICLR 2022), which fuses lifting-scheme wavelet subbands with Transformer blocks for 3D shape representation, and Wavelet-based Positional Representation for Long Context (ICLR 2025), which incorporate for Transformers.
These examples alone show that “incorporating wavelets into LLM” is not unprecedented, so the novelty claim should be carefully repositioned.

**(3) Clarity and presentation issues.** Sections 3 and 4 are largely presented as single paragraphs, which makes the paper difficult to follow. I recommend restructuring with subsections, shorter paragraphs, and (where appropriate) bullet points to highlight key design choices, algorithms, and experimental settings. In addition, several citations are missing or are written as plain text rather than using proper citation commands (e.g., \cite{}), which does not meet typical ICLR formatting standards.

**(4) Incorrect and unreliable references/URLs.** Some references appear to be erroneous. For example, at L538 the entry “Fernando Flores-Mangas. Discrete waveelet transform” contains a spelling error (“waveelet”) and appears to misattribute or misidentify the source; I could not verify this citation. Please carefully audit all references and URLs for correctness and completeness. I checked the URL provided but was unable to access it.

>Fernando Flores-Mangas. Discrete waveelet transform. The Washington Post, Spring 2014. URL https://www.cs.toronto.edu/˜mangas/teaching/320/slides/CSC320L11.pdf. (L538)

It appears Fernando Flores-Mangas did not actually write the document titled "Discrete wavelet transform".  And he doesn't seem to be a researcher in wavelet transform.

https://scholar.google.com/citations?user=jJDpzNQAAAAJ&hl=en

While I can overlook minor typos in the main text, spelling mistakes in citations or references to non-existent papers are plainly unacceptable; such errors should not occur. In the main text, Flores-Mangas (2014) is cited in the Figure 2 caption.
I knew about wavelet transforms, so I immediately noticed this citation was wrong. The author's name is clearly incorrect.
Did the author not check their own paper and fail to notice such a mistake?
**Given that the Discrete Wavelet Transform constitutes the paper’s core contribution, it is deeply concerning that its citation is wrong.**

**Furthermore, the authors mention Haar wavelet in their paper but fail to cite it in the references.** If they had properly studied wavelet transforms, such an error would be unthinkable. (To draw an analogy in LLM research, it would be like writing an LLM paper but either misquoting “Attention is All You Need” or failing to cite it altogether.)

**I remain concerned whether the author truly understands the mechanism and background of the wavelet transform.**

----
The idea may have merit, but the current submission has serious issues in experimental validation, scholarship, and exposition. Including modern baselines, thoroughly situating the work within prior research, correcting the bibliography, and improving the organization would be necessary before the paper can be considered for acceptance. Moreover, improper or inaccurate citations cast doubt on the authors’ reliability as researchers. As it stands, I cannot recommend acceptance.

**Questions:**

- Have you reviewed the ICLR Author Guidelines and used the official template for this submission?
https://iclr.cc/Conferences/2026/AuthorGuide

- Have you read Ten Lectures on Wavelets by Baroness Ingrid Daubechies? It is a foundational text on wavelet theory, and I strongly recommend studying it first.

- Have you also rread ICLR-accepted papers that include “wavelet” in the title? Using their paragraph structure, citation practices, and exposition as guides would improve the manuscript.

---

### Official Review · Reviewer_pUqz · 2025-10-28

**Soundness:** 1
**Presentation:** 1
**Contribution:** 1
**Rating:** 0
**Confidence:** 2

**Summary:**

This paper proposes a novel idea to integrate traditional wavelet transformations in signal processing into the LLMs, which speeds up the LLM pre-training without adding extra parameters to the model.

**Strengths:**

In terms of originality, the authors designed a novel module called WaveletGPT that can be inserted between the decoder layers. The module modifies the intermediate embeddings using the wavelet transformations.

In terms of significance, the evaluations show that this new approach improves the convergence rate and the negative log likelihood (NLL) scores across multiple tasks.

**Weaknesses:**

I am not familiar with wavelets, but I believe the presentation of this paper is far below the ICLR standard. The presentation is so poor that I cannot understand the details of the paper.

1. Many symbols are not sufficiently defined or are hard to find their definition. For example, I do not know if the symbol $x[n]$ on Line 141 is a $n$-dimensional vector or a vector depending on the time $n$, as $n$ is defined (much later on Line 146) as time or the context length. The letters $g$ and $h$ on Line 150 are not defined.

2. Algorithm 1 is not clear. For example, I do not know why $i < E/2$ and $i \ge \frac{E}{2}$ are put on the right side and what the algorithm is trying to do. This is not the usual way to write pseudocode.

3. Figures are repeated and inconsistent. For example, Figures 1 and 2 have a lot of overlap. And in Figure 2, the levels in the left subplot are in Roman numerals, whereas the levels in the right subplot are in Arabic numerals. This numbering difference is also presented in the main text of the paper.

4. The evaluation results in Figure 5 (left) lack explanation. I do not know how the Base, Ours, SPE, Gain, and Time are actually computed, what they represent, and whether larger is better or smaller is better. And treating the table as a figure is also inappropriate.

5. There are misused math symbols. For example, on Line 140, $[0 - 128)$ should be $[0, 128)$. On Line 262, $h(.)$ should be $h (\cdot)$

6. The citations violate the ICLR formatting instructions. For example, on Line 39, "(Nix, 2024)" should be put without the parenthesis, and `\citet{}` should be used instead of `\citep{}`. On Line 300, "Yu et al. (2023)" should be put into the parenthesis with `\citep{}`. Also, the hyperlinks do not work at all. Please check Section 4.1 of the "Formatting Instructions for ICLR 2026 Conference Submissions" document in the provided LaTeX template.

7. The grammar mistakes and typos are so many that they severely affect the reading experience, e.g., Line 134 "theycan". I cannot list all of these typos, and I suggest the authors carefully proofread their paper and fix their language.

8. The placeholder appendix is not removed from the template.

9. The sentences generally lack smooth and logical connections.

**Questions:**

What is an academic setup? What and how many GPUs are used? Do the conclusions generalize to non-academic setups?

**Details Of Ethics Concerns:**

It is highly likely that LLMs are heavily used to generate this paper without disclosure, which violates ICLR’s Code of Ethics. There are several potential LLM hallucinations in this paper.

1. Referring to non-existing equations:
There are multiple places in the paper referring to Equation 2/3/4, e.g., Line 160, but there are no such equations.

2. Fabricated citation (Line 538):

> Fernando Flores-Mangas. Discrete waveelet transform. The Washington Post, Spring 2014. URL https://www.cs.toronto.edu/~mangas/teaching/320/slides/CSC320L11.pdf.

 This citation is suspicious in the following aspects:
- There is a typo in the title: waveelet.
- The content is unlikely to be published in a daily newspaper like the Washington Post, and a daily newspaper is not published seasonally in Spring 2024.
- The URL links to the slides of a course at the University of Toronto and is not a proper reference.

3. Markdown grammar in LaTeX (Lines 204-205):

> \*approximate signal\*

This is a Markdown grammar for the Italic font, which is likely to be copied from an LLM's web interface.

---

> ### Author Response · Authors · 2025-11-27
> **Concerns about the quality of the review**
>
> I strongly protest the quality of the reviewer and flag the paper for research integrity issues.
>
> ICLR guidelines clearly mention that LLMs can be used for polishing, edits of grammar, checking for citations and everything else.
> We have made the disclosure too in the submission form stating "Yes, LLMs are used to aid or polish writing. "
>
> The quality of the review is extremely poor as the reviewer is not qualified for the review as it is mentioned and I quote "I am not familiar with wavelets". That being said, if there are grammatical errors, it is right to point them but not a ground for giving 0 out of 10. This is certainly not a english writing conference.
>
> Reviewer: "It is highly likely that LLMs are heavily used to generate this paper without disclosure, which violates ICLR’s Code of Ethics. There are several potential LLM hallucinations in this paper. "
> We have disclosed this. In addition one is allowed to cite the source and we think it is fair and ethical to do so be it slides on the internet. Similar is the case for the Washington Post article.
>
> For these reasons we declined to be part of any discussions regarding this paper, and refused to be part of the rebuttal process. I hope in future the program chairs take into account the quality and proper allocation of the reviewers.

---

> > ### Comment · Reviewer_pUqz · 2025-11-27
> >
> > Dear authors,
> >
> > For your information, the ICLR policies on Large Language Model usage can be found at this link: https://blog.iclr.cc/2025/08/26/policies-on-large-language-model-usage-at-iclr-2026/
> >
> > Regarding the submission form:
> > - The "Large Language Models" entry is not currently visible to reviewers.
> > - The disclosure choice has a second sentence, which is ignored in the authors' last response.
> > The full statement is "Yes, to aid or polish writing. **Details are described in the paper.**"
> > I do not find the relevant discussion in the paper.
> >
> > Regarding the score:
> > - There are multiple reasons to give the "0: strong reject." rating.
> > The grammar mistakes are not the only ones.
> > All reasons are listed in the review.
> > I will increase my rating if all issues are properly resolved.
> > - The rating and the confidence are scored separately in the reviewer's form.
> >
> > Regarding the potential LLM hallucinations:
> > - As reviewer `79tD` also points out, the citations indeed contain false facts.
> > - Please check the policies in the link above. Notice the following information:
> > > Ultimately the paper's authors are responsible for the contents of their submissions. Consequently, a substantial falsehood, instance of plagiarism, or misrepresentation produced by an LLM would be considered a Code of Ethics violation on the part of the paper's authors.
> >
> > As a reviewer, I only flag the potential issues. I leave the ultimate recognition of whether there is an ethics problem to the ICLR ethics team and the ACs.
> >
> > I wish the authors a good success in their current and future submissions.

---

### Official Review · Reviewer_6iEf · 2025-11-01

**Soundness:** 3
**Presentation:** 2
**Contribution:** 2
**Rating:** 4
**Confidence:** 4

**Summary:**

This paper proposes WaveletGPT, a novel approach that combines wavelets into LLM architecture. They modify intermediate embeddings of the decoder blocks with wavelet coefficients, incorporating multi-scale structure. In order not to disrupt the embeddings, they keep the level of wavelet coefficients fixed for an embedding coordinate. The Language models incorporated with these modifications converge faster than the baseline unstructured LLM across multiple modalities.

**Strengths:**

1. They propose a novel way of using wavelets for pretraining language models
2. Empirical evaluations show that their model can achieve the performance of a GPT-like architecture around 40-85% faster
3. They have an option to train the wavelets and they show that this improves the model's performance.

**Weaknesses:**

1. The benefits of using Wavelets in Language models is studied from an empirical standpoint without theoretical formulation and analysis.
2. The experiments conducted were broad and on smaller-scale models without evidence of how the architecture scales to larger sizes
3. The downstream performance of the language models and the reasoning capabilities are not studied in the paper.
4. An ablation study on the wavelet kernel size function along the embedding dimension will help understand its impact better.
5. Does the idea work for image/video generation?

**Questions:**

Please see the weaknesses

---

### Official Review · Reviewer_t1Xb · 2025-11-01

**Soundness:** 2
**Presentation:** 2
**Contribution:** 2
**Rating:** 4
**Confidence:** 3

**Summary:**

The paper inserts a "causal multi-scale convolution/averaging" module grouped by embedding dimension between standard decoder layers: half of the dimensions retain the original resolution, while the other half are smoothed using a window that increases with dimension, allowing the same layer to see both fine-grained and coarse-grained context simultaneously. The authors pre-trained or modeled it on text, raw/symbolic music, acoustic tokens (LibriSpeech), and LRA, concluding that it converges to the same loss faster (approximately 40–65% epoch saving) with almost no increase in parameters.

**Strengths:**

1.The idea is simple and straightforward, easy to integrate into existing decoders, and requires no changes to the attention mechanism.

2.Multimodal experiments demonstrate that this inductive bias does not only affect one tokenization method.

3.The learnable convolutional kernel version provides a variant that is "not purely handmade Haar," indicating that the structure is learnable.

**Weaknesses:**

1. The model is clearly outdated, and the comparison scope is too narrow. The main experiments are still based on GPT-2, short contexts, early LM datasets, and small-scale audio sets, which cannot prove effectiveness for modern autoregressive models in 2025. There is a lack of comparison with modern LLM configurations. I understand the difficulties of experimentation under resource constraints, but if the goal is to shift the focus away from scaling, as claimed in the authors' abstract, it should at least adapt to some newer strategies and architectures, and demonstrate the potential for this method to achieve superior results when combined with scaling.
2.The current manuscript reads as a sequence of loosely connected experiments and clarifications. Sections are long and under-segmented; method, extensions, and dataset-specific details are interleaved. This makes the paper harder to follow than it needs to be.
3.Your current appendix section is a template. You need to delete it if you don't use it.
4.These are narrow downstream targets compared to what is typically used to assess LLMs (QA, world knowledge, etc.) in text domain. In current LLM practice, pre-training and post-/instruction-tuning are tightly coupled: the value of a pre-training modification is determined by whether it produces a better starting point for those downstream stages. A faster reduction of language-modeling loss on small or homogeneous corpora, even together with modest gains on FSD-50K or LRA, does not by itself establish that the resulting checkpoint will transfer better to mainstream LLM downstream workloads.

**Questions:**

See above.

---

### Meta-Review · Area_Chair_LNFD · 2026-01-04

**Summary:**

The paper proposes a lightweight, wavelet-inspired multi-scale modification to decoder embeddings to accelerate convergence in small GPT-style models. While reviewers acknowledge the idea is intuitive and show some empirical speedup under limited compute, the weaknesses outweigh the strengths. The experiments rely on outdated, small-scale baselines, novelty is insufficiently contextualized with respect to existing wavelet–Transformer work, and the manuscript suffers from serious presentation, organization, and referencing issues. Taken together, these issues significantly limit the technical and scholarly contribution of the work, and the recommended decision is to reject.

**Reviewer Concerns:**

Most key concerns remain unresolved, including outdated small-scale baselines, overstated novelty due to insufficient positioning, and serious presentation and referencing issues. The rebuttal responds to only one of four reviewers and does not resolve the raised concerns. Additionally, LLM usage is indicated in the submission form but not disclosed in the paper or appendix as required by ICLR policy, which further weighs against acceptance.

**Reviewer Scores:**

The paper receives one strong reject, one reject, and two borderline reject scores. Based on the rebuttal and limited discussion, reviewer assessments would likely remain unchanged.

---

### Decision · Program_Chairs · 2026-01-26

Reject